# LiNi_0.6_Co_0.2_Mn_0.2_O_2_ Cathode-Solid Electrolyte Interfacial Behavior Characterization Using Novel Method Adopting Microcavity Electrode

**DOI:** 10.3390/molecules28083537

**Published:** 2023-04-17

**Authors:** Rahul S. Ingole, Rajesh Rajagopal, Orynbassar Mukhan, Sung-Soo Kim, Kwang-Sun Ryu

**Affiliations:** 1Graduate School of Energy Science and Technology, Chungnam National University, Yuseong-gu, Daejeon 34134, Republic of Korea; 2Department of Chemistry, University of Ulsan, Doowang-dong, Nam-gu, Ulsan 44776, Republic of Korea

**Keywords:** solid electrolyte, microcavity electrode, interfacial analysis, ionic conductivity, solid-state battery

## Abstract

Due to the limitations of organic liquid electrolytes, current development is towards high performance all-solid-state lithium batteries (ASSLBs). For high performance ASSLBs, the most crucial is the high ion-conducting solid electrolyte (SE), with a focus on interface analysis between SE and active materials. In the current study, we successfully synthesized the high ion-conductive argyrodite-type (Li_6_PS_5_Cl) solid electrolyte, which has 4.8 mS cm^−1^ conductivity at room temperature. Additionally, the present study suggests the quantitative analysis of interfaces in ASSLBs. The measured initial discharge capacity of a single particle confined in a microcavity electrode was 1.05 nAh for LiNi_0.6_Co_0.2_Mn_0.2_O_2_ (NCM622)-Li_6_PS_5_Cl solid electrolyte materials. The initial cycle result shows the irreversible nature of active material due to the formation of the solid electrolyte interphase (SEI) layer on the surface of the active particle; further second and third cycles demonstrate high reversibility and good stability. Furthermore, the electrochemical kinetic parameters were calculated through the Tafel plot analysis. From the Tafel plot, it is seen that asymmetry increases gradually at high discharge currents and depths, which rise asymmetricity due to the increasing of the conduction barrier. However, the electrochemical parameters confirm the increasing conduction barrier with increased charge transfer resistance.

## 1. Introduction

The global markets for electric vehicles (EVs) and hybrid electric vehicles (HEVs) were the fastest growing sectors from the last decade due to the supportive policies and concerns in relation to global warming. There have been ~7 million EVs and HEVs on roads globally in the last decade, and the number will increase very sharply, with it predicted to rise to ~1.4 billion in the next 10 years [1]. Among them, almost all the current EVs and HEVs are powered by Li-based batteries (LBBs) such as the Li-metal battery, Li-ion battery (LIB), Li-sulfur battery, and fuel cells such as alkaline fuel cells (AFCs). Moreover, the current LIBs and AFCs are very expensive, have less energy density to power the modern EVs [2,3], and LIBs are used liquid organic electrolytes which raise serious safety concerns (nowadays, behind every LIB explosion the flammable liquid organic electrolyte is a major concern [4]) and may leak-out of the electrolyte [5,6]. Due to these concerns, all-solid-state lithium batteries (ASSLBs) with non-flammable inorganic solid electrolytes (SEs) are receiving maximum attention from global researchers and the market. So, intensive efforts have been made by researchers in recent decades to develop high performance ASSLBs with high ionic conductive solid electrolytes.

Specifically, ASSLBs are proposed as the safest batteries due to their non-flammable inorganic solid electrolyte. Among the variety of inorganic solid electrolytes, sulfide-based inorganic solid electrolytes exhibit high ionic conductivity, mechanical stability, and wide operating temperatures. More specifically, argyrodite-based Li_2_S–P_2_S_5_–LiX-type solid electrocytes exhibit high ionic conductivity value which is comparable with organic liquid electrolytes [7,8,9]. Apart from good characteristics, sulfide-based solid electrolyte-based ASSLBs need more development to achieve better performance and commercialization. Particularly, the electrode–electrolyte interface should be improved without any side reactions [10,11]. Thus, it is necessary to study the electrochemical performance and lithium–ion migration through various interfaces of ASSLBs using various characterization techniques. For example, when ASSLBs are discharged, Li-ions have been known to migrate through the various interfaces, such as the anode–solid electrolyte, solid electrolyte–SEI (solid electrolyte interface) layer, and SEI–bulk cathode interface, in order to transfer from the anode to the cathode via the solid-state electrolyte [12]. Unfortunately, the imperfect analysis of interfaces may be hampering the electrochemical performance analysis of ASSLBs. To understand interfacial characteristics in detail and improve the performance of ASSLBs, a quantitative analysis of interfaces is required by using various analysis techniques and tools. However, micro-electrochemistry is the most powerful method of analyzing the electrochemical parameters of active particle–solid electrolyte materials such as interfacial kinetics, mass transport [13,14], and equilibrium potential using a microcavity electrode system. The microcavity electrode is partially modified from previously used single particle electrodes (Appendix A) [15] to overcome the issues regarding the special optical microscope arrangement, specific size and shape of the active particle, and contact loss, etc.

In the current research, a microcavity electrode was used in order to investigate the kinetics and the real electrochemical parameters at the interface between NCM622-Li_6_PS_5_Cl solid electrolytes along with a pulse measurement technique. The cavity electrode system is adopted for an analysis of the real electrochemical properties of active particles and electrolytes confined in the cavity to exclude the effects of surrounding interfaces, barriers, and side reactions caused by battery components around the electrodes and also the impact of the loading and current collectors of the composite electrode. In this cavity electrode measurement, it is possible to measure quasi-equilibrium potential by applying a rest time at the corresponding discharge depth. Moreover, in each depth of discharge (DOD) state, the charging and discharging are repeated due to applied various currents, and the resulting charge–discharge over-potential is plotted to obtain a Tafel plot. For example, in the case of measurement at 10% DOD, only 10% of the capacity is discharged after the material is fully charged, and the quasi-equilibrium potential is measured for the rest time in that state so that the correct equilibrium potential can be confirmed. The accurate equilibrium potential measured through the rest time via the pulse measurement and the Tafel plot obtained from the overvoltage according to various charging and discharging currents can increase the reliability of the electrochemical parameters obtained from the calculations.

## 2. Results and Discussion

The Li_6_PS_5_Cl solid electrolyte was synthesized using high energy ball milling subsequently sintering at 550 °C as shown in Figure 1. The crystal structure and the powder X-ray diffraction pattern of the prepared Li_6_PS_5_Cl solid electrolyte are shown in Figure 1a,b, respectively. The powder XRD analysis confirmed the formation of well-crystalline material after the sintering process at 550 °C for 10 h. Further, we have observed the characteristic peaks at 2θ = 15.5°, 17.9°, 25.5°, 29.9°, 31.3°, 45.0°, 47.9°, and 52.4° for the prepared Li_6_PS_5_Cl solid electrolyte. The observed diffraction peak positions were well matched with the cubic crystal structure from the F-43m space group [16]. This confirms the successful formation of a Li-argyrodite-type solid electrolyte.

In addition, we also performed the laser Raman analysis to study the structural units of the prepared solid electrolytes, and the corresponding spectrum is shown in Figure 1c. The Li_6_PS_5_Cl solid electrolyte exhibits the characteristic peaks at 179.9 cm^−1^, 280 cm^−1^, and ~420 cm^−1^, indicating the successful formation of PS_4_^3−^ structures; this arose due to the stretching vibration of tetrahedral PS_4_^3−^ anion and confirmed the formation of Li_6_PS_5_Cl argyrodite. The FE-SEM analysis showed that the prepared solid electrolytes particle size is in the range of 5 to 10 µm and has an interconnected porous granular morphology; the corresponding image is shown in Figure 1d. The ionic conductivity of the prepared solid electrolyte is obtained from electrochemical impedance spectroscopy analysis. The ionic conductivity of the prepared Li_6_PS_5_Cl solid electrolyte is calculated from the Nyquist plot (Figure 2a). The calculated ionic conductivity of the solid electrolyte is 4.8 mS cm^−1^ at room temperature. In order to calculate the activation energy, we have measured the ionic conductivity of the prepared Li_6_PS_5_Cl solid electrolytes at 30 °C to 100 °C, and the corresponding Arrhenius plot is shown in Figure 2b [17]. The calculated activation energy of the Li_6_PS_5_Cl solid electrolytes is 0.29 eV.

The initial charging and discharging analysis of the NCM622-Li_6_PS_5_Cl solid electrolyte system was conducted for the initial three cycles at a constant current density of 0.2 nA using a microcavity electrode. The initial charge–discharge and differential analysis results of the microcavity electrode cell are shown in Figure 3a,b. In Figure 3a, a typical voltage curve [18] of the positive electrode NCM622 is shown, and a phase transformation was seen according to the lithium–ion intercalation and deintercalation. For the conformation of the phase transitions of the NCM622 cathodes during Li intercalation and deintercalation, the differential voltage profile was calculated by taking derivatives of the initial charge−discharge curves, as shown in Figure 3b.

The charge–discharge analysis of the NCM622-Li_6_PS_5_Cl solid electrolyte system was measured through a microcavity measurement; the irreversible charge/discharge results were shown in the formation section of the first cycle, which are directly related to the SEI formation and phase transition; after that, with increasing cycles the coulombic efficiency was recovered and achieved 96% after the third cycles. The second and third cycles confirm the reversibility and stability of the active material. In addition, the NCM622 particle showed a discharge capacity of 1.05 nAh in the Li_6_PS_5_Cl solid electrolyte. The electrochemical parameters were evaluated from the calculated values, and the calculation was carried out considering that the shape of the cathode particle was spherical. When the lithium intercalation/deintercalation reactions are controlled by the charge transfer process, the relationship between the applied current density (i) and the generated over-potential (η) was explained by applying the Butler–Volmer equation [19], known as the Tafel analysis.

Figure 4 and Appendix A show the Tafel plots of the NCM622-Li_6_PS_5_Cl system at 10% DOD to the 90% DOD with a 20% step width; in Figure 4 only 10% DOD and 50% DOD are shown; Appendix A shows the 30%, 70%, and 90% DOD. The microcavity electrode is used to quantitively evaluate the transfer parameters by confining the active material and electrolyte in the microcavity. The results obtained from the Tafel plots of the present study are shown in Figure 4, and the electrochemical kinetic parameters were calculated. An exchange current density (*i*_0_) was estimated by measuring points forming a straight line with a slope of −*αF*/2.3*RT* (=Tafel line) [20].



(1)
$$\log i=\log i_{0}-\frac{\alpha F}{2.303RT}\eta$$



Here, *i* is the applied pulse current, *i*_0_ is the exchange current density, *α* is the charge transfer coefficient for the anodic and cathodic reactions and can be obtained from the slope of the linear line, *F* is the Faraday coefficient, *R* is the gas constant, *T* is the temperature, and *η* (=*E* − *E*_OCP_) is the overpotential value (*E* is the measured potential, and *E*_OCP_ is the open circuit potential). The charge transfer resistance *R*_CT_ at the interface and the exchange current density can be correlated as follows:(2)$$R_{CT}=\frac{RT}{Fi_{0}}$$

Figure 4 shows the relationship between the overpotential and the logarithm of applied current density with the Tafel fittings (the α = 0.8); this figure also shows that at lower discharge current densities, there is linear behavior, and with increasing discharge currents there is a large deviation from the linearity because of Li-ion intercalation through the interface being mostly controlled by the interface resistance [21]. The large deviation increasing gradually is seen for the discharge currents with the DOD increasing from 10% to 90%; thus, these asymmetricities rise due to increasing the conduction barrier between the active material and the electrolyte. This is consistent with the energy barrier seen with ionic conduction [19,22]; the presence of an asymmetrical energy barrier for a simple charge-transfer reaction can be attributed to the intrinsic electronic properties of solid electrolytes [22]. The transference fraction in solid electrolytes is only due to the cation diffusion; thus, the interfacial atomic structures could be significantly disordered and form defectively [23].

Additionally, asymmetry during the Li insertion/desertion means that the transfer of lithium ions does not occur smoothly at the beginning and end of the discharge, and this result is confirmed by the quantitatively calculated values of the charge transfer resistance and exchange current. This suggests that the electrode kinetics for the NCM particle varies with the Li concentration in the solid electrolyte. Figure 5a shows the measured values of charge transfer resistance and exchange current at various DOD states, while Figure 5b shows the diffusion coefficient values under the assumption that Fick’s diffusion equations [24] were followed using the values obtained through the pulse method. Appendix A shows the applied pulse of various current densities in each DOD state to the NCM622-Li_6_PS_5_Cl solid electrolyte system. The charge transfer resistance obtained from the measurement and calculation showed an initially high value, with an increase in DOD of up to 30% showing the lowest value; the charge transfer value increases with increasing DOD. The initially high charge transfer resistance is due to the internal phase transformation, and after 30% DOD and with increasing DOD, the charge transfer resistance also increases due to the structural instability of the active material [18]. In addition, it was also confirmed that the solid electrolyte showed a large difference in charge transfer resistance depending on the amount of lithium; the diffusion coefficient also correlates with these results. At 30% DOD, the charge transfer resistance showed the lowest value and the exchange current showed the highest value; at that stage, the Tafel plot showed the most symmetrical behavior. Table 1 compares the electrochemical parameters of previously reported results of NCM electrodes using various methods and current calculated results using the cavity electrode with pulse method; the table also shows the current results and reports the real analysis of kinetic properties. Finally, it was confirmed that the diffusion coefficient, exchange current, and charge transfer resistance were analogues to each other.

## 3. Experimental Details

### 3.1. Preparation Method of Solid Electrolyte

The lithium argyrodite (Li_6_PS_5_Cl) solid electrolyte was prepared via the high-energy ball mill process using Li_2_S (99.98%, Sigma Aldrich, St. Louis, MI, USA), P_2_S_5_ (99%, Sigma Aldrich), and LiCl (≥99%, Sigma Aldrich) as starting materials. Firstly, Li_2_S, P_2_S_5_, and LiCl were mixed with in a molar ratio of 5:1:2. Then, they were ground for 15 min using a mortar and pestle and transferred to an 80 mL planetary ball mill alumina vessel with 25 zirconia balls (10-mm diameter) were added and sealed. The alumina vessel was brought out from the glove box and mounted on the ball mill machine (Pulverisette, Fritsch, Markt Einersheim, Germany) and the balls were milled for 15 h at 400 rpm. During the ball milling process, the rotation direction was changed every 30 min after 10 min of rest to avoid overheating and enhance the uniform mixing/milling. After the ball milling process, the product was collected and ground for 15 min to get fine amorphous solid electrolyte powders. Finally, the amorphous solid electrolyte was heat-treated at 550 °C for 10 h with a heat rate of 2 °C min^−1^. The obtained solid electrolyte was named as Li_6_PS_5_Cl.

### 3.2. Materials Characterizations

The crystalline structure of the solid electrolyte material was studied using a Rigaku-Ultima (IV) X-ray diffractometer (XRD) using Cu-Ka radiation with the wavelength of 1.5418 Å. The XRD experiment was performed within the 2θ of 10 to 70° with a 0.01s step size. To identify the structural units of the synthesized argyrodite solid electrolyte, we have performed the Laser Raman spectroscopy analysis (Thermo Scientific, DXR Raman instrument, Waltham, MA, USA) with a power of 8 mW and an excitation wavelength of 532 nm. The particle size and the surface morphology of the synthesized solid electrolytes were studied using a field-emission scanning electron microscope (FE-SEM, JSM-7610F, JEOL Ltd., Tokyo, Japan).

### 3.3. Electrochemical Characterizations

The ionic conductivity of the solid electrolyte was measured via electrochemical impedance spectroscopy (EIS) analysis using the Biologic SP 300 instrument with a frequency range of 7 MHz and 1 Hz at room temperature (25 °C). For this EIS analysis, the solid electrolyte (250 mg) was pressed into a pellet with a 10-mm diameter and ~2.2-mm thickness using hydraulic pressure (35 MPa). Before pressing into the pellet, a 50-µm thick indium foil was attached on both sides to ensure good contact, and then it was assembled in a pressure cell (Appendix A).

### 3.4. Microcavity Electrode Preparation and Characterization

For the preparation of the microcavity electrode system, initially prepare the gum-like electrode by mixing the NCM622 powder of spherical particles with sizes of ~12 µm (confirmed with the microscope, Appendix A shows the FE-SEM image of the NCM622 powder) as a synthesized argyrodite-type solid electrolyte (Li_6_PS_5_Cl) and conductive material (Super P) and poly(tetrafluoroethylene) (PTFE) binder in the ratio of 80:18.5:1.5:1 wt.%. The as-prepared electrodes were used to characterize the NCM622 secondary particles and solid electrolyte interfacial analysis in the corresponding systems.

The electrochemical parameters of the secondary particle NCM622 cathode material and solid electrolyte were investigated by using the microcavity electrode measurement. After the active cathode material particles were inserted into the microcavity electrode, a solid electrolyte was also pushed into the cavity for efficient contact with the electrolyte. A lithium metal foil was used as the counter electrode. The charge/discharge analysis of secondary particles was performed using an IVIUM (Iviumstate, Ivium Technologies, Eindhoven, The Netherlands) in the potential range of 3.0–4.3 V vs. Li/Li+ at room temperature. Initially, the electrode was cycled three times in the CC mode for conformation of the reversibility and stability of the particles; after that, it was subjected to one full charge, then the discharge process was divided according to the DOD state (DOD, 10–90%). In each DOD step, charging and discharging were performed at various current densities using the pulse method.

## 4. Conclusions

We successfully prepared the high ion-conductive Li_6_PS_5_Cl argyrodite solid electrolyte using high-energy ball milling. The structural study shows that the prepared electrolyte had a pure argyrodite cubic crystal structure with a F-43m space group, while the surface morphology shows that the interconnected porous grains had grain sizes between 5–10 nm. The ionic conductivity measurement gives a high ionic conductivity and is 4.8 mS cm^−1^ of the Li_6_PS_5_Cl argyrodite-type solid electrolyte (at room temperature). Moreover, the microelectrode analysis was successfully characterized using a modified microcavity electrode and a prepared solid electrolyte. As a result of the quantitative evaluation of the electrochemical properties of NCM622 in the Li_6_PS_5_Cl solid electrolyte using the microcavity electrode, the measured initial reversible capacity was 1.05 nAh. The Tafel plot confirms the maximum symmetry of charge transfer at the 30% DOD state; also, this state shows the lowest charge transfer resistance and highest exchange current and diffusion of Li-ion. The high ionic conductivity and quantitative electrochemical parameters make the Li_6_PS_5_Cl solid electrolyte an excellent candidate for next-generation ASSLBs.

## Data Availability

The datasets used and/or analyzed during the current study are available from the corresponding authors at a reasonable request.

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
