# Peer review of "LiNi0.6Co0.2Mn0.2O2 Cathode-Solid Electrolyte Interfacial Behavior Characterization Using Novel Method Adopting Microcavity Electrode"

_molecules, 2023, doi:10.3390/molecules28083537_

Round 1

Reviewer 1 Report

1. The authors are suggested to revise the introductions section significantly. several new aspects in renewable energy technologies should be added for fair discussion. For example: (https://pubs.acs.org/doi/full/10.1021/acsami.2c21814, Chemical Engineering Journal 454, 140289)

2. The authors are requested to provide details of charge transfer resistance calculation and fitting curves.

3. A scheme for the preparation method could be added.

Author Response

  1. The authors are suggested to revise the introductions section significantly. several new aspects in renewable energy technologies should be added for fair discussion. For example: (https://pubs.acs.org/doi/full/10.1021/acsami.2c21814, Chemical Engineering Journal 454, 140289)

Response: Thank you for your suggestions. We have revised the Introduction part considering the renewable energy technologies (hydrogen fuel cells).

  1. The authors are requested to provide details of charge transfer resistance calculation and fitting curves.

Response: Thank you for your kind comment. We have added more detailed information about charge transfer resistance and fitting curves (lines 157 – 170 in the manuscript). The revisions look as follows:

“An exchange current density (i0) was estimated by measuring points forming a straight line with a slope of – αF/2.3RT (= Tafel line) [19].

log i = log i0η                                                          (1)

Here, i is the applied pulse current, i0 is the exchange current density, α is the charge transfer coefficient for the anodic and cathodic reactions and can be obtained from the slope of the linear line, F is the Faraday coefficient, R is the gas constant, T is the temperature, and η (= EEOCP) is the overpotential value (E is the measured potential, and EOCP is the open-circuit potential). The charge transfer resistance RCT at the interface and the exchange current density can be correlated as follows:

RCT =                                                                        (2)”

  1. A scheme for the preparation method could be added.

Response: Authors thank the reviewer for the suggestion, we have added the schematic diagram in the revised manuscript.

Reviewer 2 Report

Paper fits the scope of the journal. The experiments were properly described. There are two suggestions:

1. Chapter 2. Results and Discussion should be after 3. Experimental Details.

2. Time and thermal stability of the material should be also discussed.

Author Response

  1. Chapter 2. Results and Discussion should be after 3. Experimental Details.

Response: Thank you for your kind comment. We refer to the journal manuscript template. According to the journal`s manuscript preparation template, the sequence of the manuscript should be:

  1. Introduction
  2. Results
  3. Discussions (Chapters 2 and 3 can be combined)
  4. Materials and Methods (Experimental part)
  5. Conclusions
  6. Patents

  1. Time and thermal stability of the material should be also discussed.

Response: Authors thank the reviewer for the valuable suggestion. Unfortunately, we don’t have the facility for high temperature thermal analysis (at inert atmosphere). Usually, argyrodite crystallization occurs after 300℃ and the same was studied in our previous work (Scripta Materialia, 204, 2021, 114129).

Reviewer 3 Report

1. The abbreviation NCM622 appears in the title and abstract, and its full name should be given. Similarly, in line 76, DOD appears and its full name should be displayed.

2. In Figure 1 ( a ), the schematic diagram of the crystal structure of the prepared Li6PS5Cl solid electrolyte, what elements occupy each lattice node need to be marked.

3. The authors claim that “ the Li6PS5Cl solid electrolyte exhibits the characteristic peaks at 179.9 cm-1, 280 cm-1, and ~ 420 cm-1. However, the characteristic peaks shown in Fig.1 ( c ) are the stretching vibration of tetrahedral PS43- anion, and this arising due to and confirmed the formation of Li3PS4 structure. It seems unclear to explain the formation of Li6PS5Cl.

Author Response

  1. The abbreviation NCM622 appears in the title and abstract, and its full name should be given. Similarly, in line 76, DOD appears and its full name should be displayed.

Response: The authors are thankful for your comment. We have checked all of the abbreviations and revised them.

  1. In Figure 1 (a), the schematic diagram of the crystal structure of the prepared Li6PS5Cl solid electrolyte, what elements occupy each lattice node need to be marked.

Response: Authors thank the reviewer for the suggestion, we have marked elements in the schematic diagram and the corresponding figure in included in the revised manuscript.

  1. The authors claim that “the Li6PS5Cl solid electrolyte exhibits the characteristic peaks at 179.9 cm-1, 280 cm-1, and ~ 420 cm-1”. However, the characteristic peaks shown in Fig.1 ( c ) are “the stretching vibration of tetrahedral PS43- anion”, and “this arising due to and confirmed the formation of Li3PS4 structure”. It seems unclear to explain the formation of Li6PS5Cl.

Response: Authors thank the reviewer for the comments, actually, Li3PS4 is the basic structure of Li6PS5Cl argyrodite, anyhow we have modified the statement in the revised manuscript as below:

“The Li6PS5Cl solid electrolyte exhibits the characteristic peaks at 179.9 cm-1, 280 cm-1 and ~ 420 cm-1, indicates the successful formation of PS43- structures and this arising due to the stretching vibration of tetrahedral PS43- anion and confirmed the formation of Li6PS5Cl argyrodite.”
